# Nurse, Give Me the News! Understanding Support for and Opposition to a COVID-19 Health Screening System

**DOI:** 10.3390/ijerph20021164

**Published:** 2023-01-09

**Authors:** Natalia Gulbransen-Diaz, Soojeong Yoo, Audrey P. Wang

**Affiliations:** 1School of Architecture, Design and Planning, The University of Sydney, Camperdown, NSW 2006, Australia; 2Wellcome/EPSRC Centre for Interventional and Surgical Sciences (WEISS), University College London, London W1W 7TY, UK; 3Biomedical Informatics and Digital Health, School of Medical Sciences, The University of Sydney, Camperdown, NSW 2006, Australia; 4DHI Laboratory, Research Education Network, Western Sydney Local Health District, Westmead Health Precinct, Westmead, NSW 2145, Australia

**Keywords:** COVID-19, self-service technology, health screening, health services, hospital, design, implementation, digital health, qualitative study, Internet of Things, Internet of Medical Things

## Abstract

Helping the sick and protecting the vulnerable has long been the credo of the health profession. In response to the coronavirus-disease-2019 (COVID-19 pandemic), hospitals and healthcare institutions have rapidly employed public health measures to mitigate patient and staff infection. This paper investigates staff and visitor responses to the COVID-19 eGate health screening system; a self-service technology (SST) which aims to protect health care workers and facilities from COVID-19. Our study evaluates the in situ deployment of the eGate, and employs a System Usability Scale (SUS) and questionnaire (*n* = 220) to understand staff and visitor’s acceptance of the eGate. In detailing the themes relevant to those who advocate for the system and those who oppose it, we contribute towards a more detailed understanding of the use and non-use of health-screening SSTs. We conclude with a series of considerations for the design of future interactive screening systems within hospitals.

## 1. Introduction

The pith of the health system is care. Whether it is the provision of care in response to an injury, to assist in the early detection of disease or to simply encourage the maintenance of good health, healthcare workers have long been caring for the sick and protecting the vulnerable. However, the novel coronavirus pandemic (COVID-19) has proven to be one of the most significant health challenges of the century. Public health approaches to containing and mitigating the pandemic include a diverse portfolio of activities: many governments have established varying degrees of border control with restrictions on both domestic and international travel [1]; a bolstering of their health system’s preparedness, readiness and response capabilities [2]; the prohibition of large-scale events and public gatherings; and the employment of surveillance measures [3]. From an individual perspective, citizens now participate in varying levels of contact-tracing, self-isolation/quarantine and have increased their engagement in hand washing, respiratory etiquette, and social distancing [4].

Various healthcare facilities specifically–as sites which house some of our most vulnerable populations–have begun to explore the implementation of syndromic surveillance systems to mitigate the spread of COVID-19 and prevent an outbreak. Syndromic surveillance refers to methods that rely on detecting the clinical health indicators of an infectious disease–such as discernible behavioural patterns, signs and symptoms [5,6,7]–before a confirmed diagnosis is made via laboratory or gold-standard testing. Operationalising these systems largely consists of re-stationing existing health workers to conduct the resource-intensive and manual screening procedure.

On the surface, self-service technologies—“technological interfaces that enable customers to produce a service independent of direct service employee involvement” [8]—appear perfectly suited to taking over the syndromic health screening process during a global pandemic. Self-service systems have the potential to offer scalable solutions, promote health and safety for frontline workers and reduce the burden of labour on healthcare staff to allow them to focus their efforts on patient-specific care rather than managing entry procedures to the hospital. However, despite this dreamlike ideal, there remains a scarcity of research exploring the application of self-service technologies for syndromic surveillance and health screening within a hospital context.

Thus, a slew of questions are proposed: (1) what aspects of automated health screening is a self-service system appropriate for? (2) what is the overall expense of implementing self-service technologies for health screening? (3) what physical and tangible form is optimal for these technologies? (4) how can the deployment and diffusion of these technologies influence staff and visitor experiences entering a hospital? (5) what is an adequate rate of technological acceptance? and (6) how might hospital administration track the value of this technology?

While this paper does not seek to answer all of these questions, it is motivated by the scarcity of knowledge in this space. This paper is contextualised within a novel COVID-19 smart Internet of Things (IoT) screening system pilot (hereafter: eGate) by an interdisciplinary research team in partnership with a metropolitan children’s hospital in Australia. We describe how the eGate was created–it’s setup and core mechanisms–and deployed in order to evaluate how hospital staff and visitors perceive and respond to these systems (particularly during periods of acute stress such as a global pandemic). The aim of this research is to investigate how self-service health screening technologies might be created in hospital environments. Our study provides an interpretive formative evaluation at the end of the pilot project [9,10] and addresses the highlighted research gap between design and implementation science [11]. Although our research will not provide answers to all of the queries previously outlined, it will, in plain terms, address the following research questions:1.To what extent do hospital staff and visitors accept the eGate as a self-service health screening system?2.What factors should designers be mindful of during the design, development and implementation of future health screening systems?

These questions offer a strong foundation for the other considerations to be addressed in the future. By investigating staff and visitors initial acceptance or rejection of the eGate, we provide some insight into the complex experience of these devices and hope to encourage future research that is capable of responding to the more nuanced questions raised.

This paper is structured accordingly: it outlines relevant knowledge regarding novel technologies in the health context, existing uses of self-service technologies for healthcare, and the distinction between use and non-use of digital technologies. It then details how and why the eGate was designed, as well as how the SUS and questionnaire data was collected and analysed. Key insights, illustrating how staff and visitors experience, accept and respond to the pilot eGate screening system are presented, alongside a comparison of insights that were of statistically significant difference between “advocates for” and “opponents of” the system. With these findings in mind, the paper shifts its focus to a broader discussion of how the eGate itself was realised, and how future technologies might be envisaged. The contribution of our work is threefold: firstly, we offer insights into the acceptance and rejection of an in situ self-service health screening system. Secondly, outline how factors influencing eGate acceptance may differ between participants who have positive experiences with the eGate and those who did not, discussing codes that have significant implications for future designs. Finally, we reflect on the opportunities and challenges raised by the eGate pilot and offer recommendations for future systems.

## 2. Background

This paper draws on insights related to innovative technologies in a health context, the use of self-service technologies to control borders, and the study of non-use with various interactive systems. Relevant research from each of those fields is introduced below.

### 2.1. Novel Technologies in the Health Context

In countries such as the United Kingdom, New Zealand and Australia, health care organisations that provide free-at-service care are often large, bureaucratic and have many competing resource priorities in non-pandemic times. These resourcing issues are further amplified in developing countries [12].

The future of public health response increasingly will require the adoption of digital tools [13,14,15,16] and integrative technologies such as Internet of Things [17,18,19]. Unfortunately, there remains a paucity of research on the real-world impact of these tools in health systems due to barriers creating slower uptake [20,21] and whilst public policies are still catching up [22,23]. Digital organisational and systemic transformation of health care has been slow to date, affecting digital maturity and adoption, and reflected in digital health implementation-related research [24,25,26,27]. The potential of these technologies remain unleveraged due to legacy information technology system issues such as interoperability, differing data formats that might prohibit ease of data sharing for rapid monitoring, follow-up and communications [23,28].

The COVID-19 pandemic has appeared to have accelerated prioritisation [29] of digitisation through sheer necessity, such as natural accelerated use of telehealth to support ongoing continuity of care that is safer for socially-distanced patients and staff—the efficacy and effectiveness of these innovations though still require further investigation [30,31].

Nonetheless, these services are essential in providing an early warning data-driven tool for monitoring signals prior to or in an ongoing crisis [5]. However, there remains a lack of published evidence on evaluating and quantifying the detection capability of multi-purpose syndromic services. Each system is often complex operationally, individualised to detecting the event such as an uptick in a specific infectious disease and previous results have varying successes [32,33]. Nevertheless, syndromic surveillance evaluations are found in far more common seasonal events such as the flu than COVID-19 [34,35] and evidently research priorities for COVID-19 sensors that could be used in these systems have only been recently outlined [29].

### 2.2. Self-Service Technologies for Health Screening

The notion of “self-service” has a much longer history than most suspect—think of making a phone call by dialling directly rather than being dependent on a telephone operator, or the ability to push buttons in elevators instead of using an elevator operator [36]. However, recent advances in information technology have also produced a number of opportunities in which self-service technologies can be leveraged for increased efficiency and convenience [37,38]. Common implementations of self-service technology in healthcare are often patient-focused. Examples include patient management self-service systems [36], which assist in patient admission and/or discharge, and online appointment systems [36,39]. Furthermore, digital technologies developed in response to COVID-19 are often designed with a singular purpose, such as diagnosis (e.g., AI tools to distinguish between novel coronavirus pneumonia and influenza pneumonia [40], a chatbot/symptom to disease health assistant which differentiates between various diseases [41]) or prevention and surveillance (e.g., the patient-facing self-triage and self-scheduling tool Coronavirus Symptom Checker [42], various forms of contact tracing [43]). To our best knowledge, there are no known examples of integrated Smart IoT systems implemented in hospitals for the purpose of COVID-19 screening.

As such, a more reliable comparison (akin to the size and function of the eGate) of self-service technology has been deployed in various airports around the globe for the purpose of automating border control via an ePassport gate. These Automated Border Control (ABCs) self-service barriers are positioned at various checkpoints throughout airports in Europe, and require travellers to complete the necessary border checks without an intermediary [44]. The core function of ePassport gates is to “authenticate the travel document, establish that the traveller is the rightful holder of the document, query border control records, and automatically verify the entry conditions” [45]. The gates primarily consist of (a) an electronic document (passport) reader, (b) biometrics readers, (c) an electric door opened by electronic means, and (d) a device to display visual instructions [45].

Furthermore, while there are similarities in the overall intentions of ABCs and the eGate, to the best of our knowledge, an ABC system is yet to be adapted to a hospital context for the intention of syndromic surveillance and COVID-19 management. We argue that the significant contextual differences between airports and hospitals means that findings are not entirely transferable.

For example, airport biometrics screening services are often concerned with ensuring a secure and streamlined travel experience for passengers–essentially verifying passenger identification processes through a range of physical properties [46]. Automated syndrome surveillance in hospitals, on the other hand, would seek to verify a visitor’s *health* status through the visitor’s presenting (but often changing) signs and symptoms. Put simply, ABCs seek to verify a single host (i.e., the individual passing through the gate), while in a hospital a health-screening system would need to verify the status of a host *and* any pathogens attached to said host (i.e., any infectious diseases or viruses).

While we believe that a self-service eGate system could mediate a hospital entrance in the same manner as the automated border control gates within airports, the eGate outlined in this research will provide insights into the real-world implementation of the technology.

### 2.3. Learning from Non-Use

We argue that examining non-use (when and how people do not use technology) is an informative line of inquiry [47], while it is indeed important to understand when and how people use technology (and who future users might be), understanding non-use and learning *“what not to do”* can be just as valuable. Wyatt explores patterns of non-use particularly with respect to those who adopt and then stop using Internet technologies, in contrast to the conventional diffusionist image of permanent adoption, and uses that to examine the policy implications for different groups [48]. Selwyn (2003) is similarly concerned with non-use, and with the ways in which traditional rhetorics of technological progress “pathologizes” non-use [49]. Those conceptualisations have significant ramifications in the conduct of HCI research and practice. For example, definitions of “user” form the implicit basis for such central concepts as “user interface”, “user study”, “user experience”, and others. Documenting non-use exposes implicit assumptions about who the user is not [50].

Prior work [34,49] also outlines how a great deal can be learned when technology is placed in the context of non-use, such as being able to see first-hand the different ways technologies may not integrate effectively into their intended environment and consequently fail [47]. HCI research can learn a good deal about technology use by placing it in the context of non-use, because when we do so, we see it not as simply an inevitable response to some inexorable march of technological progress, but rather as a creative, complex, and contingent act of its own.

In line with work from Samaras et al. [34] and Selwyn [49], staff and visitor’s use and non-use of eGate health-screening system will provide much insight into how effectively the pilot technology was implemented in the hospital context, and how long it may remain there with the approval of staff and visitors. Exploring non-use also provides valuable suggestions for future iterations of the technology and redesigns of the system.

## 3. Study Design

The eGate pilot (Figure 1) is a self-service health screening system that aims to improve health system resilience and pandemic readiness. Figure 1 and Figure 2 illustrate the three major automated aspects: (1) the dynamic display and communication of COVID-19 case locations, (2) user temperature check and health screening, and (3) distribution of the daily sticker that is always displayed when inside the hospital.

### 3.1. Need for the eGate

Health screening protocols, prior to the eGate pilot, required groups of up to six concierge screening staff to be stationed outside each of the hospital’s entrances in order to control staff and visitor’s access to the hospital. This manual screening process required the concierge screening staff to: (1) physically screen staff and visitors face-to-face, (2) ask entrants a series of risk assessment questions, (3) take the entrant’s temperature and ensure it was within an acceptable range.

Screening questions were changed regularly based on updated state government’s health advice, but generally pertained to whether or not the entrant had been to any of the recent COVID-19 case locations or exposed public transit routes. Additional questions regarding any flu-like symptoms (based on the World Health Organisation’s (WHO) COVID-19 protocols) or if they had been in contact with anyone who had tested positive to COVID-19 in the previous 14 days. Concierge screening staff were able to assess the entrant’s temperature with a handheld infrared forehead thermometer (temperature < 37.4 °C/99.32 °F to pass). If the individual was determined by concierge screening staff as being healthy and passing the test, the concierge screening staff would give the person a coloured sticker (the colour changed daily) to affix on their clothes. The sticker was used to signify that an individual had been screened for the day, allowing them to leave and enter as they wished.

The core challenges with this method, however, is that it is incredibly labour-intensive, requiring manual temperature checks and risk-assessment questions, and required concierge screening staff to be drawn from pre-existing healthcare staff teams. At a time when medical resources are already stretched, a practical and less human resource-intensive solution to support the public health response was suggested, allowing more healthcare workers to return to the provision of direct patient care and minimise potential infectious exposure of staff and the public to COVID-19.

### 3.2. eGate Mechanisms

The eGate health screening system (Figure 2) uses a custom COVID-19 Screening web-app and three different physical interfaces for COVID-19 automated screening (the physical eGate, a customised app reader and temperature sensor). The COVID-19 Screening App uses evidence-based COVID-19 screening questions and also integrates an API from the state government, detailing case locations and transport routes impacted by COVID-19 cases, which finally generates a QR permission token code for physical gate-enabled access, accessed via the QR token custom reader. The QR token data is checked and validated in real time with the temperature checks via a thermal camera sensor, before the eGate is opened. Privacy preserving algorithms are used to store data in the cloud servers using the state based health WiFi access points.

The pilot system has been trialled by more than 1500 staff and regular visitors during the 9-month pilot in 2021. A single concierge screening staff member was located alongside the eGate for maintenance and additional entry assistance. Vetted senior staff can access the database if necessary, for digital contact screening and create public announcements via the app which is easily seen by any staff member coming into the hospital daily. The app also displayed individualised user friendly feedback about temperature readings and communicated the latest information on COVID-19 symptoms and locations of concern. The app has additional anticipated capability to store other types of validated access control information if necessary such as verification of COVID-19 vaccination status.

### 3.3. Data Collection

Data were collected over a one-month period (19 August 2021 to 21 September 2021). During this period, 43,484 individual interactions were recorded with the eGate in order to enter the hospital, reaching an average of 1450 people per day. After individuals had completed their online health screening survey (in order to generate the necessary QR token) and passed through the eGate, they were then presented with the optional research survey to share their experiences with the health screening system. Consent to participate in the research was indicated through the completion of the optional survey post ethical approval.

The embedded research survey contained a System Usability Scale (SUS) evaluation and additional open-ended, experience oriented questions. SUS is one of the most widely-used standardised questionnaire assessments of perceived usability [51] and a recognisably reliable and robust method for measuring a participant’s ability to complete a task using a system [52]. The three additional open-ended questions were formulated in order to shift the participant’s focus beyond the strict *usability* of the system (explored via the SUS evaluation) to a more experiential lens [53], and aimed to understand what aspects of the system participant’s liked, what they would improve and any other comments they would like to share. As many participants were completing the questionnaire as they entered the hospital to begin their working shift, a quick assessment of their experience was prioritised without compromising their opportunity to express feelings, impressions, and attitudes regarding the system [54].

### 3.4. Data Analysis

To find and meaningfully report patterns within our data, we analysed the questionnaire responses using an inductive thematic analysis in NVivo 12 [55]. Of the 220 individuals who completed the SUS, 59.09% (*n* = 130) responded to the first question, 64.09% (*n* = 141) responded to the second, and 43.18% (*n* = 90) responded to the third question.

Analysis of these responses was performed by two of the authors in three stages to abide by Gioia’s three orders of coding [56]. During the first stage of coding all questionnaire responses were read and first order codes were identified. Code labels were intentionally similar to the participant’s own terms in order to better represent their own lived experience [56]. 309 raw data nodes (i.e., participant statements) were consolidated into 29 1st-order codes. In the second stage of coding we sought to identify similarities and differences so that the 28 germane codes could be abstracted into seven 2nd-order themes–each underpinned by their theoretical connotations. The final stage of coding investigated the potential of any further thematic distillation, and produced four aggregate dimensions: (1) interaction as dialogue, (2) enhancing user skill set, (3) implementation of novel technology, and (4) ideological consternation.

Our study employed an across-stage mixed methods approach to further explore how usability factors (identified in the thematic analysis) might differ between users and non-users. We used the mean of the SUS evaluations (57.5) to separate our sample into two groups. Participants who scored the eGate above 57.5 are classified as “advocates of the system”, while those who scored the eGate below 57.5 are described as “opponents to the system”. While the use of quantitative methods on qualitative results is uncommon, we argue that the practice is particularly useful in this context as our dependent variable (usability factors defined by participants’ own lived experience) is entirely subjective and phenomenological [57]. de Groot distinguishes between “hypothesis testing research”, whereby a researcher seeks to determine if predictions derived from precisely postulated hypotheses are accurate, and “material-exploration”, whereby a researcher’s expectations about associations present in the data, are neither precise nor postulated in advance [58]. de Groot (2014) further clarifies that the goal of material exploration is to “let the material speak” through ad hoc processing decisions [58]. Importantly, we acknowledge that this ad hoc approach precludes the exact interpretability of our statistical outcomes, but nonetheless, argue that we are not trying to argue for evidential impact. Rather, we offer an interpretation of results which may be useful for future hypothesis-generation and subsequent hypothesis testing research.

To this end, we utilised a *t*-test to compare the prevalence of 1st-order codes in the results of “advocates” and that of the “opponents”. Prevalence of codes (i.e., whether a code is present in the participant’s questionnaire response) was prioritised over frequency of codes (i.e., the number of times a code appears) to control for participants’ verbosity, while 1st-order codes were selected for the level of specificity of phenomena which they describe. A Benjamini–Hochberg or a false discovery rate procedure was also applied on the resultant *p*-values to control for Type 1 errors (false positives) [59]. Supporting information is available in the Appendix A.

## 4. Results

220 individuals (Table 1) entering the hospital and passing through the eGate completed a System Usability Scale (SUS) evaluation [52].

The results of this research are presented in three parts. Section 4.1 details the mean SUS score (55.7) given to the eGate and details the distribution of scores across the entire sample. Section 4.2 outlines the four usability factors that encompass staff and visitor’s experiences with the eGate. Finally, Section 4.3 combines insights from the prior two sections to offer insights into staff and visitor acceptance and rejection of the eGate system. This is done by exploring user experiences (thematic codes) that are significantly more prevalent in either “*advocates of the system*” or “*opponents to the system*”.

### 4.1. SUS Results

Current studies demonstrate that a mean SUS score of 68 is considered to be average, with scores above 68 considered as above average, and vice versa for those below 68 [60]. The 220 SUS evaluations were converted to a uniform score ranging from 0 to 100, with a mean score of 55.7. This correlates with a D SUS grade for the eGate system, describing it as an “okay” system that is marginally acceptable [60].

Table 2 details the division of scores across the entire sample. At the top of the scoring range, 30 participants (13.6%) rated the eGate between 84.1 and 100, correlating to an A+ grade or the “best imaginable” descriptor. The other 6 participants who gave the system an A grade had scores ranging between 78.9 and 84.

14 participants (6.36%) scored the system a B (72.6–78.8), and 34 participants (15.45%) gave it a C (62.7–72.5). Notably, 61.8% of participants gave the eGate a SUS score within the D or F range, correlating to the “OK”, “poor” and “worst imaginable” descriptors.

### 4.2. Identified Usability Themes of the eGate Health Screening System

Our qualitative analysis identified four key themes: *interactions as dialogue*; *enhancing user skill set*, *implementation of novel technology*; and, *ideological consternation*. These 3rd-order themes capture key aspects of staff and visitor’s acceptance of the eGate health screening system. Table 3 presents an overview of our analysis.

Interaction as dialogue reflects the complex human-system relationship, as well as the various nuances that influence this dynamic. Put simply, references in this theme compare the human-system interaction to a dialogue between two beings, in which “listening” (i.e., perceiving and understanding the other party) is just as important as “speaking” (i.e., putting forth interpretable, useful and necessary information).

Enhancing user skill set refers to processes and functions embedded within the digital, self-service system that reinforce the user’s role as an active and participating being. The other side of the proverbial coin to *interaction as dialogue*, this theme captures the processes through which the system is able to encourage the user towards correct and efficient use during interactions, as well as aspects of the system that are either designed to assist the user generally, or designed specifically to increase the knowledge, capability or autonomy of the user.

Implementation of novel technology outlines the pragmatic, tangible and communication practices which are essential to distributing new processes into a hospital context. Deeply entwined with the context in which the design is situated this theme reflects concerns raised by hospital staff regarding the concept’s integration into the hospital environment, as well as various hardware reliability issues and their consequences. The “accuracy” dimension of information provided to the system was also amalgamated in this theme, with participant’s referencing the shift in power from manual screening (in which a nurse or staff collects the screening data) to a self-service system (whereby users are responsible for the accuracy of their information) and the implications of this transition.

Ideological consternation is the frame of thinking through which participants’ broader engagement with the eGate is interpreted through. Either situated at a level of thinking above or simply beyond their immediate physical and tangible interaction with the system, this theme highlights the ways in which an individual’s broader mentality (or ideology) relates to their experience of the system. Participant’s who expressed an ideological consternation almost always framed said consternation as the priority of their feedback. A predetermined disagreement with the design concept coloured their interactions with the eGate–perhaps to the point of overlooking any benefits the system may, or may attempt to, offer. We believe this notion would stand vice versa, whereby a participant’s predetermined approval of the system might blind them to any negatives, however, no participant in our sample demonstrated a predetermined, positive stance.

### 4.3. Understanding Acceptance and Rejection of the eGate System

Six of the 29 first-level codes were found to be significantly distinct after corrections: *time consuming*, *deployment communication*, *reliability of collected data*, *necessity of the system*, *lack of a feedback loop*, and *retaining the humanity of staff*. That is to say, while 23 first-level codes were equally prevalent amongst the “advocates for the system” and “opponents to the system”, the aforementioned six first-level codes were significantly *more prevalent* and reflect some of the most significant considerations and complications of the eGate system.

#### 4.3.1. Time Consuming (Greater Prevalence in Opponents to the System, *p* = 0.0053)

Time consuming is a code within the ‘physical and tangible complications’ theme, part of ‘implementation of novel technology’. Mentioned three times by the advocates of the system and 22 times by those opposed to it, this code was a clear point of delineation. Those frustrated with the system argued that *“13 pages to complete for a 1 off appointment is ridiculous! I work in health and other areas have nothing as convoluted as this” (P193)* and *“A single ‘I know and understand the restrictions’ tick box rather than a 7 min survey that wastes thousands of hours of our lives (collectively)” (P163)*. Other participant quotes in this code focused specifically on the questions asked, and suggested they were *“Necessary questions although tedious” (P66)*. It is perhaps worth noting that these concerns regarding the timeliness of engagement with the system can be juxtaposed with another 1st-order code *quick engagement with the system*. This code appeared seven times within the data set (*n* = 5 advocates of the system, *n* = 2 opponents of the system) and contained quotes such as *“quick to get through” (P16)* and *“faster than the old screening process” (P81)*. Nevertheless, the greater prevalence of the *time consuming* code (25 appearances rather than 7) indicates that future health screening systems may face an uphill battle to be accepted if they are not as fast as, if not faster than the previous screening method.

#### 4.3.2. Deployment Communication (Greater Prevalence in Opponents to the System, *p* = 0.0067)

Deployment communication is a code within the ‘diffusion and deployment’ theme, part of ’implementation of novel technology’. Participant’s who opposed the system expressed a frustration with the system that was closely linked to their perceived lack of training: *“there was no warning to arrive early to take the time to register” (P)* and *“most people still seem not to know how to download it into the phone” (P193)*. Another participant argued that *“to do it properly it would take 30 mins–it needs to be sent out and completed before arrival” (P213)*. Overall, the sentiment of these participants demonstrates the importance of concentrated efforts to communicate all relevant deployment communications (e.g., justification of concept deployment, coverage of concept progression, appropriate preparations needed for ease of use, etc.).

#### 4.3.3. Reliability of Collected Data (Greater Prevalence of Opponents to the System, *p* = 0.0067)

Reliability of collected data is a code within the ‘authority of information’ theme, also part of ‘implementation of novel technology’. Concerns regarding the accuracy and efficacy of the data given to the system were expressed more frequently by participant’s who opposed the eGate system, while participant’s who advocated for the system rarely mentioned data reliability (*n* = 1), quotes such as *“if someone is actually reading the full case locations while just entering the hospital at work, then I would be really surprised” (P194)*, *“I watched people who were filling out the form, scroll straight to the bottom, without reading the list of suburbs” (P29)* and *“I can guarantee nobody is looking through all those routes/venues” (P213)* communicate doubt in the overall effectiveness of the system. Furthermore, the tone of participants’ quotes suggests that doubts regarding the reliability of the data (i.e., a cornerstone of the health screening system) may be easily exacerbated into doubts regarding the necessity for the system at all. This was exemplified by one participant who wrote, *“[the eGate] has prolonged the process of entering the hospital and it uses a less accurate temperature screening process” (P219)*. Doubts regarding the legitimacy of a system’s working parts–both digital and physical–may easily become justification for ideological consternation.

#### 4.3.4. Necessity for the System (Greater Prevalence in Opponents to the System, *p* = 0.0192)

Necessity of the system is part of ‘ideological consternation’ dimension, and demonstrates that while some participant’s had concerns with the reliability of the system (surmised in *implementation of novel technology*), other staff members did not agree with the *premise* of the concept itself: *“an updated count of how many cases the app has effectively turned away from the hospital. Although it’s such a useless app I imagine the number is still zero 2 months into this wave.” (P156)*, *“this is useless nonsense that will have no improvement in COVID outcomes. It’s a waste of government resources to make the admin staff who do not have any real understanding of clinical practise feel good about their COVID responses” (P183)*. Hospital staff also commented on the cost of the project, stating that they *“generally thought it was too expensive” (P43)*. This code presents a significant challenge to technological acceptance, as participant’s fundamentally opposed to (or simply unaware of) the need for the system are challenging to convert into users.

#### 4.3.5. Lack of a Feedback Loop (Greater Prevalence of Opponents to the System, *p* = 0.0346)

Lack of a feedback loop is a code within the ‘diffusion and deployment’ theme, part of ‘implementation of novel technology’. Frequently referenced in relation to various physical or tangible errors present in the system itself, participants felt poorly equipped to either report or self-manage errors. One participant asked for *“better instructions—a go to in person (an actual person) kiosk rather than trying to troubleshoot at concierge and hold everyone up” (P148)* while another requested *“some technical support or help page or similar… when there are issues there is no information about where to get help” (P35)*. Importantly though, the notion of feedback did extend beyond just physical and tangible complications and encompassed a general distaste for having their voice heard in the design, development and deployment of the proof of concept. This is exemplified by one participant writing *“Only that there is the feedback option to tell the team how useless the app is” (P156)*.

#### 4.3.6. Retaining the Humanity of Staff (Greater Prevalence in Opponent’s to the System, *p* = 0.0378)

Retaining the humanity of staff is also part of the ’ideological consternation’ dimension. This code specifically referenced how hospital staff perceived to be valued by higher administration (not valued at all), as well as how they interpreted their role in the hospital (merely a cog in the machine). On the first point, a participant mentioned that while they *“fully support the technology and the screening process…the requirement to go through gates coming into work sends a terrible message to staff and is inordinately insulting” (P107)*. Another participant compared the mechanical gate to a *“prison” (P181)*. To the latter point, staff mentioned *“it is dehumanising to be identified by a QR code” (P119)* when they are doing very real, valuable and essential work caring for and protecting vulnerable individuals.


*“We forget from the top down of the organisation that crisis dehumanises all of us. From patients who become COVID pending/COVID positive/COVID negative to staff who are moved around like chess pieces to fill gaps in rosters to exec who are scrambling to keep up and keep going. I am not sure barcoding us does anything but reinforce that. Surely the [entry] sticker could say something to make staff and families feel heard and encouraged. We all feel depleted and like small squashed cogs in a large complex beast—barcodes reinforce our namelessness, the fact that we are expendable and unknown.”*
(P148)

## 5. Discussion

Having clarified how our participant’s responded to the eGate, we would now like to shift the perspective of this paper towards our second research question: what factors should designers be mindful of during the design, development and implementation of future health screening systems?

We hope that this broader reflection on how the eGate was realised will help to inform how future technologies of this kind may also be realised. Leveraging our learnings regarding the hospital staff and visitor’s acceptance and rejection of the system, these recommendations are framed through a speculative alternative scenario which illustrates where and how these learnings may be situated in the design process. Despite it is marginal application in this context, scenario-based thought experiments have been crucial in public administration, social action and decision-making more broadly [61].

In line with the authors’ design experience, the alternative design process is structured according to the four core components of design innovation in health and medicine: insight, intent, design, and action [62]. This reflects the iterative, and at times ambiguous, nature that is central to design practice (Figure 3). It acknowledges that while project planners may desire systematic and integrated approaches to digital health implementation and evaluation, more rapid, iterative and flexible approaches are often required.

Within medical design innovation, insight focuses on establishing the context of the design scenario and collection of the information that inspires the design, while the intent component serves as an opportunity for the project team to articulate the intended outcome and impact of the design [62].

Design captures the various design, scaffolding and prototyping activities which, through iterations, form and improve the design. Action pertains to the implementation and evaluation aspects of a design project, while we are framing our speculations through the lens of design innovation in medicine and health [62], we believe our recommendations are generalisable enough to be applied across most design processes.

To that end, Figure 3 positions the qualitative themes identified in the usability and acceptance evaluation of the eGate, and positions them alongside the respective component of innovation in health and medicine within which they fall. Additional relationships between the four components, further highlighting how aspects of each theme have impact across multiple stages of the design process are included. Questions capturing the core purpose of each stage are also listed.

### 5.1. Understanding Ideology (Insight)

Rethinking the *insight* component of the eGate design project suggests that a greater understanding of the needs, values and wants of the stakeholders was needed. This is particularly apparent when considering the needs, values and wants of the healthcare staff who have to engage with the eGate everyday–often multiple times per day. The necessary and repeated engagement with the system delineates the needs of hospital staff from that of visitors, and this distinction in needs has significant implications, while training and regular use of the eGate do improve the user’s experience through confidence and understanding (*Once I got used to it… it was ok.” (P)*), current comments about the eGate demonstrate the importance of a positive first impression. Many of the participant’s who expressed strong frustration for the eGate commonly did so through the theme of *ideological consternation*, expressing a fundamental opposition to the *idea* of a health screening system. Earlier engagement with these users—or framing hospital staff as stakeholders–would be a viable way of understanding these experiences and frustrations within the broader hospital context, and informed design concepts that attended to these experiences appropriately. Design frameworks such as co-design and participatory design, which afford users an equal role in the design and development of products, services and experiences, would be particularly appropriate [63,64].

In addition to expanding the core insights included, we suggest that a further understanding of organisational constraints could better inform how the design is integrated into the hospital context. Hospitals are complex institutions with diverse existing structures and cultural mores that any health screening system must be cognisant of for seamless integration. Context mapping methods [65] could further illuminate existing work cultures operating within the hospital and provide essential information regarding prior processes of technological acceptance. From these approaches, design and implementation teams may better equip themselves with the knowledge and understanding needed to develop concepts that are cognisant of their organisational or structural limitations.

### 5.2. Mitigating Consternation (Intent)

As previously described, the intent component of a design project clarifies the intended outcome of the project’s efforts. In layman’s terms, it is often the stage when a design “brief” is written and the specific project requirements and outcomes are articulated. Despite significant expressed frustration from users, we still believe the eGate achieves the originally outlined intent: syndromic surveillance *is* an effective method for mitigating disease spread. Self-service technologies *have* operated effectively in the context of border control before. The technology involved *does have* the capacity to be easily scaled, controls for entry can easily be altered in response to a dynamic medical advice, and the system unshackles hospital staff from manually operating a resource-intensive screening process.

Nevertheless, ideological consternation arose and many of the participants questioned both the necessity of the system and why the system portrayed them as disposable. Despite the technical viability and operational feasibility of the eGate, staff and visitors responses to this project highlight how the eGate falls short in meeting a desirable and usable experience for users.

Explicit engagement with users as stakeholders (proposed in Section 5.1) should extend to the articulation of intent as well to ensure that the outcome of the project truly aligns with the needs, wants and values of all stakeholders. Future self-service health screening systems, and additional iterations of the eGate, should not neglect the very human component of this system and instead capitalise on the opportunities offered. The eGate is frequently the first touchpoint of the hospital that staff and visitors interact with as they seek care or begin their work, and this offers a unique opportunity to offer empathy and comfort to visitors in vulnerable positions or health care staff who do their best to support these individuals.

### 5.3. Testing Human-System Interactions (Design)

During the design component, our focus lies in the significance of iterative testing and user inclusion. Many of the issues within the eGate, which exacerbated negative experiences, were often reliability or hardware issues that are not difficult to address or amend. These issues (e.g., the lack of shortcuts within the system, challenging presentation of COVID locations, ergonomic challenges of the thermal scanner) need resourcing, time, and testing to improve, while these issues did not interfere with data collection directly, an unfortunate hardware issue sometimes meant that the eGate was stuck in the “open” position. In these instances, the concierge screening staff confirmed individual’s entering had an approved QR token and passed the temperature screening as they came through the open eGate. Ultimately, additional iterative testing, enabled by clear avenues for feedback from users, could make a marked improvement on the system.

We encourage any future self-service health screening services to be particularly conscious of how the human-system interaction is framed. Design evaluation in this stage should consider the specific functions and facilities of the eGate’s hardware and software, while also exploring how the gate is experienced by visitors and staff members. Through these avenues, future systems should seek to equip (both single interaction and repeat) users with the necessary knowledge, skills and conveniences for an enjoyable and usable experience. Similarly, future eGates must also be designed as interpretable systems that clearly communicate with their users.

### 5.4. Implementing for Acceptance (Action)

The *implementation of novel technologies* theme notably contained four codes that were significantly more prevalent in responses from those who did not positively score the system. These codes demonstrate that the deployment challenges of the eGate were significant to the users. Concerns regarding the reliability of collected data, the lack of opportunities for staff to provide feedback or seek assistance, and minimal deployment communication coloured the participant’s perception of the system and could often be exacerbated to a fundamental opposition of the eGate concept. From this learning, we stress the importance of ensuring that the implementation of self-service screening technologies is handled with care. Ample opportunities for feedback (which is visibly actioned in some capacity) should be offered, and concentrated efforts to communicate the deployment and implementation of the system should be made.

## 6. Limitations

The nature of the eGate system (a proof of concept deployed in situ for pilot testing and evaluation) still contained a number of hardware and software bugs, and had restricted institutional resources, that served to limit the accuracy of our research. These issues, which often resulted in the mechanical gate getting stuck in either an open or closed position or the thermal scanner being inopportunely positioned, had a significant impact on the usability experience of hospital staff and visitors, while this offered us an understanding of visitor and staff’s initial responses to an early iteration of this technology, additional evaluations of self-service health screening systems ought to be conducted in order to further understand the environmental and contextual aspects of implementation. Future iterations of the eGate system may also provide more insight into the broader impact of these systems (i.e., the extent to which they reduce wait times at entry points, the validity of user-supplied data, etc.).

## 7. Conclusions

Mitigation and containment have been the mainstay approaches of managing the high transmissibility of COVID-19 [66]. As the global pandemic has progressed, governments and their hospitals rapidly explored the potential of technology to facilitate pandemic management activities that are difficult to achieve manually [67]. Self-service technologies are uniquely positioned to facilitate health screening services at entry points of hospitals, akin to Automated Border Control systems which augment immigration processes and border-crossing points at airports. Yet, as previously outlined in our introduction, a slew of unanswered questions remain and there is no–to the best of our knowledge–rigorous exploration of *how* exactly these systems might operate, *nor* how hospital staff and visitors may accept or reject these systems.

Our research leveraged the piloting of the eGate and a qualitative, design-led evaluation of said system, to better understand the efficacy of the technology. Our research presented a standardised usability assessment of the eGate, four themes related to staff and visitor’s acceptance and rejection of the system, as well various codes which are of significant difference amongst participants who opposed the system. Finally, and most relevantly, we contribute a thought-based scenario experiment which reflects on the eGate pilot and presents reasoned recommendations for the design of future health screening systems through an alternative design process.

As we acknowledged at the start of this paper, this work suggests that advancing a more open field of interdisciplinary research will encourage greater investigation into the design, implementation and use of self-service technologies for health screening. We are also intrigued by potentially alternative forms of health screening. Currently, the ABCs and the eGate are both large mechanical devices that appear intimidating to users. Much like Doug Dietz’s transformation [68] of the MRI machine–from something fearful and anxiety inducing to something adventurous and wondrous for children–we are eager to explore how future health screening systems may offer moments of reassurance or support to hospital visitors and staff–all the while supporting our pandemic management activities and keeping our most vulnerable communities safe. Future research should explore how these collaborative concepts can be sustained or have long-term acceptance in the healthcare context especially for digital health transformation projects.

## Figures and Tables

**Figure 1 ijerph-20-01164-f001:**
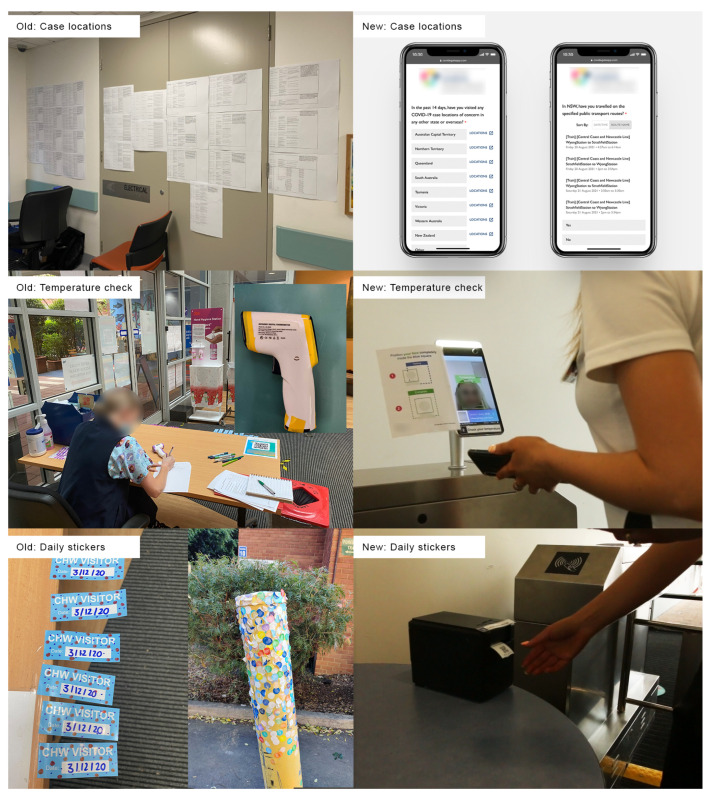
Comparing the manual system with the eGate system.

**Figure 2 ijerph-20-01164-f002:**
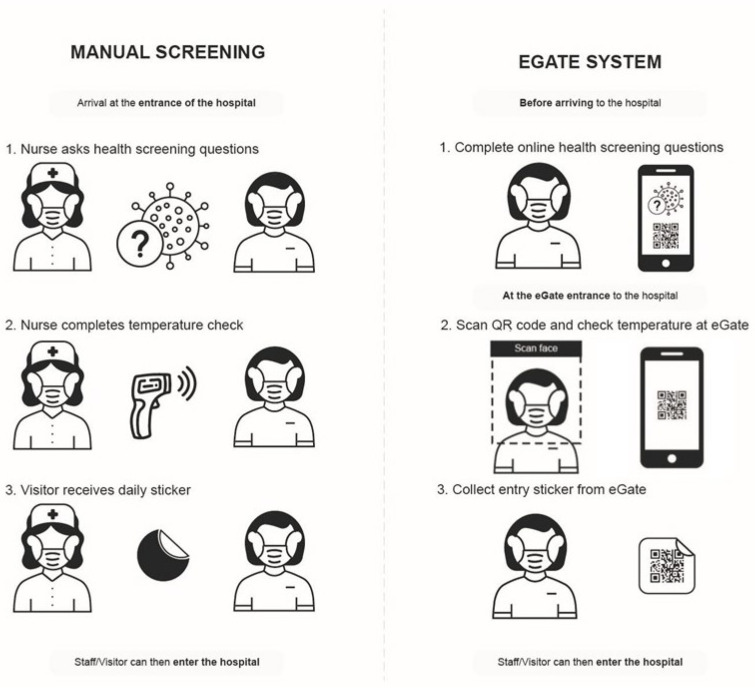
Change in agency from manual to eGate screening. **Left**: Manual screening methods; **Right**: eGate digital screening system.

**Figure 3 ijerph-20-01164-f003:**
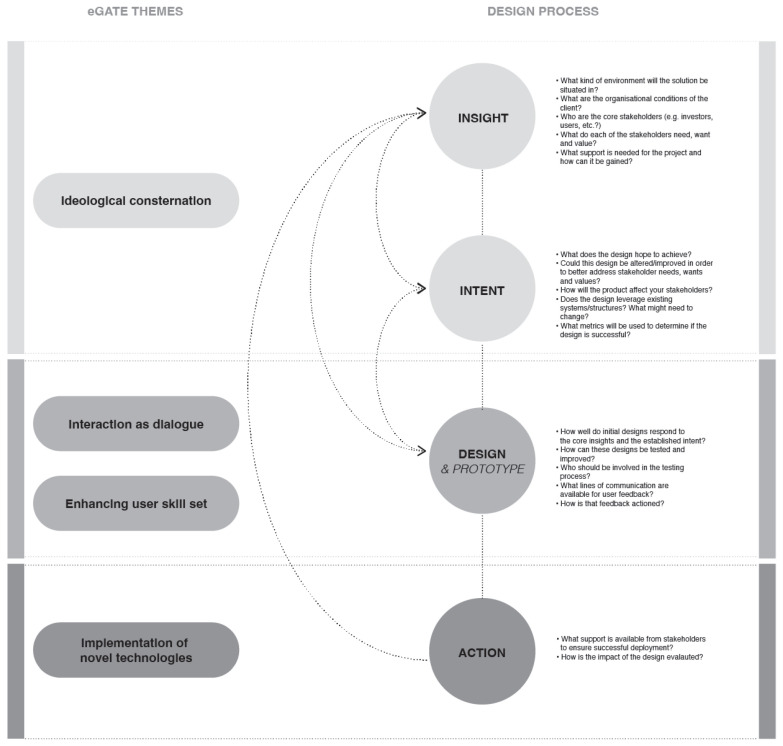
Alternative design process, inspired by the design innovation in health and medicine framework [62].

**Table 1 ijerph-20-01164-t001:** Relevant demographic information for the participants in our study.

Variable	Item	Number (%)
Gender	Female	159 (73%)
	Male	57 (26%)
	Indeterminate/Intersex/Unspecified	3 (0.01%)
Entry reason	Staff	192 (87%)
	Visitor	28 (13%)

**Table 2 ijerph-20-01164-t002:** Graded System Usability Scale score.

Grade	Grade	# by Grade	% by Grade
A	78.9–100	36	16.36%
B	72.6–78.8	14	6.36%
C	62.7–72.5	34	15.45%
D	51.7–62.6	46	20.91%
F	0–51.6	90	40.91%

**Table 3 ijerph-20-01164-t003:** Summary of the thematic analysis of staff and visitor’s experience with the eGate.

Themes	Sub-Themes	Description
Interaction as dialogue	User comprehension of service	Staff or visitor’s capacity to “read” the system correctly (i.e., understand how to interact) as they encounter it.
	Information usability	Extent to which the information communicated by the system is interpretable, useful and necessary for the user.
Enhancing user skill set	Developing user competency	Processes through which the system is able to encourage the user towards correct and efficient use during human-system interactions.
	Encouraging user convenience	Aspects of the system designed to assist the user generally, or specifically serve to increase the knowledge, capability or autonomy of the user.
Implementation of novel technology	Diffusion and deployment	Considerations and concerns raised by users that refer specifically to the ways in which the system was integrated into the hospital environment.
	Physical and tangible complications	Reliability issues with the system’s hardware and their physical and tangible consequences.
	Authority of information	The “accuracy” dimension of information provided to the system. Reflects the shift in power from manual screening (in which a nurse or staff collects the screening data) to a self-service system (whereby users are responsible for the accuracy of their information) and the implications of this transition.
Ideological consternation *		** Theme contains no sub-themes, only codes*

## Data Availability

The data that support the findings of this study are available on request from the corresponding author, A P Wang. The data are not publicly available due to restrictions, e.g., their containing information that could compromise the privacy of research participants. The study’s pre-registration can be found at https://osf.io/tpxur accessed on 22 December 2022.

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
