# Peer review of "Nurse, Give Me the News! Understanding Support for and Opposition to a COVID-19 Health Screening System"

_ijerph, 2023, doi:10.3390/ijerph20021164_

Round 1
Reviewer 1 Report
This is a report on the adoption of the eGate Screening system. The presentation on this case study is thorough and insightful regarding the usability and challenges encountered in the system application. The paper could be strengthened in two major areas as follows:
1. Clarification on the purpose of the designed system: Is this a pilot study or implementation science study?
2. Future directions: Numerous barriers are noted in the design and implementation of the eGate system. The study could employ a logic model to illustrate varying stages of the evaluation research on the e-Gate system.
Reviewer 2 Report
This paper aims at investigating staff and visitor responses to the COVID-19 eGate health screening system which is a self-service technology(SST) for protecting health care workers and facilities from COVID-19. This paper designed a system and used a System Usability Scale and questionnaire to understand staff and visitor’s acceptance of the system. This paper is timely, interesting and well-written. I have two comments as follows.
1. This study proposed a new health technology for hospital staffs. But, what is the innovation of this technology compared with the existing similar technologies?
2. This study adopted a survey to test the user acceptance of their system. However, the survey design and measurement development was not clear and convincing. Moreover, the data analysis and results are not well-presented.
Author Response
Please see the attachment and additional supplementary file is available from editor. Thank you.
